# Advancing Immunization Coverage and Equity: A Structured Synthesis of Pro-Equity Strategies in 61 Gavi-Supported Countries

**DOI:** 10.3390/vaccines11010191

**Published:** 2023-01-16

**Authors:** Vesela Ivanova, A. S. M. Shahabuddin, Alyssa Sharkey, Mira Johri

**Affiliations:** 1Public Health Agency of Canada, Ottawa, ON K1A 0K9, Canada; 2Primary Health Care and Health Systems Strengthening (PHC-HSS) Unit, MNCAH Programme, UNICEF Headquarters, New York, NY 10017, USA; 3School of Public and International Affairs, Princeton University, Princeton, NJ 08544, USA; 4Centre de Recherche du Centre Hospitalier de l’Université de Montréal (CRCHUM), Montréal, QC H2X 0A9, Canada; 5Départment de Gestion, d’Évaluation, et de Politique de Santé, École de Santé Publique (ESPUM), Université de Montréal, Montréal, QC H3N 1X9, Canada

**Keywords:** immunization, children, equity, gender, rural, remote, conflict, urban, poor

## Abstract

Background: Global immunization inequities persist, reflected in the 25 million underimmunized and 18 million zero-dose children in 2021. To identify country approaches to reach underimmunized and zero-dose children, we undertook a structured synthesis of pro-equity strategies across 61 countries receiving programmatic support from Gavi, the Vaccine Alliance. Methods: We extracted data from 174 Country Joint Appraisals and Multi-Stakeholder Dialogue reports (2016–2020). We identified strategies via a targeted keyword search, informed by a determinants of immunization coverage framework. Strategies were synthesized into themes consolidated from UNICEF’s Journey to Health and Immunization (JTHI) and the Global Routine Immunization Strategies and Practices (GRISP) frameworks. Results: We found 607 unique strategies across 61 countries and 24 themes. Strategies to improve care at the point of service (44%); to improve knowledge, awareness and beliefs (25%); and to address preparation, cost and effort barriers (13%) were common. Fewer strategies targeted experience of care (8%), intent, (7%) and after-service (3%). We also identified strategies addressing gender-related barriers to immunization and targeting specific types of communities. Conclusions: We summarize the range of pro-equity immunization strategies employed in Gavi-supported countries and interpret them thematically. Findings are incorporated into a searchable database which can be used to inform equity-driven immunization programs, policies and decision-making which target underimmunized and zero-dose communities.

## 1. Introduction

Despite tremendous immunization progress in recent decades, global inequities persist, reflected in the 25 million underimmunized and 18 million zero-dose children in 2021 [1]. Zero-dose children are defined by the Immunisation Agenda 2030 consortium (IA2030) as those that lack access to or are never reached by routine immunization services, measured by the lack of the first dose of diphtheria-tetanus-pertussis containing vaccine (DTP1) [2]. Health inequities are significant drivers of the gaps in immunization coverage and challenges in reaching the most under-served communities in low- and middle-income countries (LMICs) [3,4]. It is estimated that half of zero-dose children reside in three key geographic contexts: urban poor areas, remote communities and conflict-affected settings [5]. Moreover, gender-related barriers to immunization create additional challenges, with indicators such as maternal education and age being significant determinants of immunization coverage [3,4,6,7].

To improve immunization coverage and equity, the World Health Organization has launched the Immunization Agenda 2030 (IA2030), with the goal of “leaving no one behind”, including a core objective to extend immunization services to reach zero-dose and underimmunized children and communities [8]. Aligned with IA2030, Gavi has launched its 5.0 Strategy with the goal of reducing the number of zero-dose children by 25% by 2025 and by 50% by 2030 [9,10]. Through several funding streams, Gavi supports immunization programs in more than 70 LMICs [11]. Gavi also supports UNICEF to conduct implementation research to improve policies and programs, identify implementation bottlenecks and promote equity in immunization delivery [12].

To understand how countries are currently addressing these gaps and where there is scope for improvement, we undertook a mapping and structured synthesis of pro-equity strategies for immunization across 61 low-income countries receiving programmatic support from Gavi, expanding on a more targeted mapping exercise by Dadari and colleagues [13]. The synthesis aimed to identify current practices and promising opportunities to reach zero-dose and underimmunized children in LMICs and to inform the creation of a searchable database to enable countries to identify potential solutions to immunization challenges and support country-level programmatic planning.

## 2. Materials and Methods

Strategy mapping followed a two-phased approach. Phase I (2019) led by Dadari and colleagues focused on 13 Gavi-supported countries (Afghanistan, Central African Republic, Chad Republic, Democratic Republic of Congo, Ethiopia, India, Kenya, Kyrgyzstan, Madagascar, Myanmar, Nigeria, Pakistan, Uganda) [13]. Phase II (expanded mapping in 2021) considered data from 48 additional Gavi-supported countries. A list of countries selected for both phases is available in Appendix A.

### 2.1. Methods for the Initial Mapping by Dadari et al.

The extraction of pro-equity strategies was guided by the development of a conceptual framework by UNICEF researchers, including the determinants of immunization coverage, key thematic areas and associated keywords. Methods for the first phase are described in detail in previous published work by Dadari et al. [13].

### 2.2. Methods for the Expanded Mapping

The expanded mapping sought to include data from 48 additional Gavi-supported countries, as well as incorporate more recent data from phase I countries, when available. We also aimed to interpret data by applying additional equity filters, considering gender-related barriers to immunization and inequities in immunization for target populations (urban poor, remote rural, conflict). Lastly, we sought to organize strategies thematically and identify opportunities to make findings accessible to countries, development partners and other interested parties

#### 2.2.1. Data Source Selection

The mapping considered Gavi Annual Country Joint Appraisals (JA) reports between 2016 and 2019, aligning with the Gavi 5.0 strategy timeline. JA reports describe country-level implementation progress and the performance of Gavi funding support [14]. Countries were selected based on eligibility for Gavi support in 2020. In view of the COVID-19 pandemic in 2020, countries did not submit JA reports. Instead, countries conducted “Multi-Stakeholder Dialogues” (MSD), which aim to convene relevant stakeholders to discuss country-level barriers to immunization and align future objectives and actions. MSD reports summarizing key findings from these sessions were included as data points for that year. JAs and MSDs were obtained from the Gavi website and the Gavi Secretariat.

A total of 126 JAs (*n* = 106) and MSDs (*n* = 20) from 48 additional countries were selected for the data extraction. Up to three most recent data points were chosen for each country, when available. Additionally, given that Syria only became eligible for Gavi support in 2019 and there were no JAs or MSDs available, a report on Syria’s National Immunization Strategy was obtained as an alternative data source.

#### 2.2.2. Data Extraction and Synthesis

Strategies were identified via a targeted keyword search, informed by the phase I conceptual framework. The data extraction method was initially calibrated by manually extracting strategies from randomly selected JAs used in phase I. After careful review of the initial mapping and JA reports, additional keywords were generated for the thematic areas to improve the identification of strategies and increase the scope of the extraction.

Data extraction was facilitated by the “text retrieval” feature of the freely available qualitative analysis software QDA Miner Lite to accommodate a large dataset and increase the reliability of results. For efficiency and convenience, keywords were searched across all JA and MSD reports simultaneously. Paragraphs where keywords appeared were analyzed for their relevance and presence of strategies corresponding to the thematic areas. An Excel extraction matrix, including countries, thematic areas and keywords, was populated with strategies.

Data extraction was performed by the first author, with supplemental cross-check for 10% of the data by a second reviewer. Independently obtained results between both reviewers were compared at multiple occasions. Differences in data extraction and strategy capture were discussed among reviewers and co-authors until a consensus was reached and appropriate adjustments or calibrations were made. Additionally, all keywords were translated in French to enable data extraction from reports available only in French. The final set of keywords used for extraction can be found in Appendix B.

Data from phase I and II were collated, yielding a total of 607 unique strategies, extracted from 174 reports and 61 countries. The thematic synthesis of pro-equity strategies was informed by the analysis led by Dadari and colleagues in Phase I, UNICEF’s Journey to Health and Immunization (JTHI) framework, and the Global Routine Immunization Strategies and Practices (GRISP) framework [13,15,16]. The JTHI framework is a tool used to identify the factors influencing different points of the immunization service delivery, including before, during, and after immunization. The framework also offers an opportunity for a more targeted identification of barriers and solutions. Table 1 outlines key dimensions related to each step of the JTHI framework.

The thematic synthesis of strategies was also informed by the Global Routine Immunization Strategies and Practices (GRISP) framework [16]. The GRISP comprehensive framework of strategies and practices for routine immunization introduces key areas of action to strengthen immunization systems and improve coverage. The framework also describes a systemic approach to address barriers by tackling four categories of actions: maximizing reach, managing the program, mobilizing people and monitoring progress [16]. Among these actions, GRISP highlights nine transformative investments, aimed to guide governments to transform immunization programs and achieve better outcomes [16].

We thematically synthesized the collated pro-equity strategies, building on the themes identified in phase I, JTHI steps and GRISP dimensions. When relevant, GRISP approaches were included in the themes. Examples of GRISP themes include the integration of immunization with other routine services, strategies to address vaccine hesitancy and misinformation and practices to build the capacity of healthcare workers. Given that some strategies were found to be relevant to more than one theme and JTHI step, we opted to include them in all relevant dimensions and themes, leading to a total of *n* = 740 data points used for the analysis. To support knowledge use, a learning tool and searchable database of strategies was created by UNICEF. The tool allows filtering of results by country, JTHI steps, health system element, relevance to key populations and the application of a gender lens [17].

## 3. Results

The number of unique strategies reported by each country ranged from a minimum of one strategy (Cambodia, Mongolia, Philippines, DPR Korea, Syria) to a maximum of twenty-seven strategies (Nigeria) (see Figure 1).

### 3.1. Strategies Addressing JTHI Steps

Strategies targeting JTHI step 4 (Point of Service, *n* = 328 [44%]), step 1 (Knowledge, Awareness and Beliefs, *n* = 181 [25%]) and step 3 (Preparation, Cost and Effort, *n* = 98 [13%]) were the most common. Countries less frequently reported strategies targeting step 5 (Experience of Care, *n* = 58 [8%]), step 2 (Intent, *n* = 54 [7%]) and step 6 (After-Service, *n* = 20 [3%]). Figure 2 illustrates the number and relative proportion of strategies by JTHI step and determinants of immunization coverage.

Moreover, nested within each step and determinant of immunization coverage, we consolidated strategies into 24 themes (see Figure 3).

Pro-equity strategies relevant to JTHI step 4 (Point of Service) targeted a variety of determinants and themes, including Utilization (*n* = 115 [16%]), Management and Coordination (*n* = 81 [11%]), Human Resources (*n* = 53 [7%]), Commodities (*n* = 49 [6%]) and Budget and Expenditures (*n* = 30 [4%]). The most common theme identified at this step was: tailoring the location of service delivery (*n* = 81 [11%]), which is related to factors influencing the utilization of services by clients. For example, in Guinea, immunization supplies and equipment were installed in private, religious and armed forces infrastructures in order to expand the supply of vaccination services for populations that are harder to reach in urban areas.

Further, strategies relevant to step 1 (Knowledge, Awareness and Beliefs) targeted the determinants Social Norms (*n* = 161 [22%]) and Human Resources (*n* = 21 [3%]). The themes most commonly identified at this step were: engaging local leaders to address misinformation and raise awareness (*n* = 84 [12%]) and use of communication strategies to address misinformation and raise awareness (*n* = 76 [10%]). For example, in Lao People’s Democratic Republic, community leaders and village health volunteers have been targeting mothers in known high-risk villages by engaging them to understand their own personal views and potential hesitancy regarding immunization. Moreover, in Indonesia, a communication campaign was developed to advocate for measles and rubella immunization through short films, SMS blast messages, art/graphic design and communication channels (Facebook, Twitter, WhatsApp, etc.).

Strategies related to step 3 (Preparation, Cost, and Effort) targeted the determinants Utilization (*n* = 69 [9%]) and Management and Coordination (*n* = 29 [4%]). Strategies at that step were most commonly related to the theme: reminders and strategies to reduce time, costs and opportunity barriers (*n* = 39 [5%]). For example, in Eritrea, mothers and caregivers were encouraged by healthcare providers to bring defaulter children to sites for vitamin A supplementation and immunization during the African Vaccination Week and Child Health and Nutrition Week activities.

Strategies used at step 5 (Experience of Care) were most often related to the theme: adjusting service delivery approach and engaging community to ensure acceptability (*n* = 44 [6%]). A strategy under this theme was implemented in Ethiopia with the use of community outreach agents to perform the community-based monitoring of children eligible for vaccinations, thus engaging community leaders and volunteers. Moreover, strategies at step 2 (Intent) were most often related to the theme: financial and non-financial incentives to improve staff motivation and performance (*n* = 41 [5%]). For example, in South Sudan, 465 vaccinators and 24 supervisors were trained in interpersonal communication (IPC) skills. Lastly, we found fewer strategies targeting step 6 (After-Service), where the most common themes were: strengthening accountability, trust and communication for mobilization (*n* = 12 [2%]), including AEFI training conducted in multiple countries (Nicaragua, Eritrea, Mozambique and Myanmar, among others).

The least frequently identified themes among all JTHI steps were as follows: health workers being from the communities they serve (*n* = 6 [1%]), taking stock of post-care successes and failures (*n* = 6 [1%]), performance rewards for healthcare workers (*n* = 2 [0.2%]) and security initiatives to allow services to happen (*n* = 1 [0.1%]).

### 3.2. Strategies Targeting Key Populations and Gender

Our approach allowed the identification of strategies targeting special populations, including the key geographical contexts identified by the Equity Reference Group for Immunization and reflected in both IA2030 and Gavi’s strategy 2021–2025: poor urban areas, remote rural communities and populations affected by conflict [3,8,9,10]. Additionally, gender was identified as a cross-cutting theme and influencing factor for immunization and was incorporated in the data interpretation process. The UNICEF searchable database allows filtering of results based on their relevance to key populations and the consideration of gender [17]. Given the volume of data, a few select examples related to the most common theme identified in the mapping (engaging local leaders to address misinformation and raise awareness) are highlighted here.

For example, in Malawi, mother care groups (village head chiefs, women volunteers) were created by Malawi Health Equity Network (MHEN) in hard-to-reach areas, urban slums and refugee camps. The groups’ activities involve defaulter tracing (door-to-door), health education, interpersonal communication and advocacy at the community level.

Additionally, in Pakistan, a prototype on immunization in slums of one union council in Lahore was developed and implemented locally. Through this approach, twelve slum health committees were established with participation from local community notables, religious leaders, teachers and local government representatives for advocacy and social mobilization among slum communities.

Moreover, in Nigeria, the Women Advocates for Vaccine Access (WAVA), a coalition of women-focused civil society organizations, was formed to advocate for increased routine immunization and sustainable vaccine financing.

## 4. Discussion

We summarize pro-equity immunization strategies employed in 61 Gavi-supported countries between 2016 and 2020, as reported in JA reports. This mapping provides an overview of what is being done to improve immunization equity and reach the most disadvantaged communities in Gavi-supported LMICs. Moreover, we offer a thematic synthesis of strategies in the form of a searchable database, informed by the JTHI and GRISP frameworks [17].

Countries are using various strategies to tackle immunization challenges at each JTHI step. While there is an abundance of strategies targeting Point of Service (step 4); Knowledge, Awareness and Beliefs (step 1); and Preparation, Cost and Effort (step 3), we found gaps in addressing immunization Intent (step 2), Experience of Care (step 5) and After-Service (step 6). Mapping strategies against the JTHI and GRISP frameworks allowed the identification of common practices among countries, as well as opportunities for strategy development and future investments to improve immunization outcomes.

Strategies focused on leveraging communication strategies and engaging local leaders to raise awareness and address misinformation were common among countries. These strategies shape knowledge, awareness and beliefs about immunization among caregivers and health workers and influence the first step of the JTHI framework. Efforts aligned with this theme also support the IA2030 strategic priority to increase commitment and demand by improving public trust and confidence, acceptance and value of vaccination, and addressing the reluctance to vaccinate [8]. Vaccine hesitancy remains a key challenge in LMICs and was identified by the WHO as one of the top global health threats in 2019 [18,19]. While the prevalence of vaccine hesitancy in LMICs can vary, prioritizing efforts to influence caregiver knowledge and confidence about the importance, safety and effectiveness of vaccines can have a significant impact on vaccine uptake globally [20,21]. However, given that the causes for vaccine hesitancy can be complex and context-specific, understanding how Gavi-supported countries are addressing these challenges can inform future efforts to increase immunization coverage and equity, particularly in countries with similar socio-cultural contexts. [16,18,19].

Strategies targeting the logistics of service delivery, such as tailoring the location of services, were also common. These include efforts to provide tailored services for hard-to-reach communities, microplanning, Reach Every District (RED)/Reaching Every Child (REC) initiatives and other “bottom-up” approaches, which have previously been supported by the literature [22,23]. Additionally, countries reported strategies focused on reminders and strategies to reduce time, costs and opportunity barriers and on conducting coverage and equity assessments (CEAs). These strategies directly address the IA2030 strategic priority of improving coverage and equity as they focus on identifying barriers to vaccination due to age, location, social and cultural and gender-related factors [8]. Moreover, prioritizing CEAs can further contribute to bridging equity gaps by promoting the identification of under-served communities, the selection and prioritization of pro-equity strategies and the monitoring of pro-equity strategies, interventions and outcomes [24].

Furthermore, countries reported a variety of strategies focused on addressing systemic elements of the immunization process, such as strengthening healthcare systems, improving information and data management systems; strengthening supply chains and logistics; and investing in the healthcare workforce. These strategies demonstrate an alignment with GRISP transformative investments, including building vaccinator capacity, modernizing vaccine supply chains and investing in information systems [16].

Opportunities for strategy development can be found in the least common themes identified: health workers are from the communities they serve and security measures to allow immunization services to happen safely in conflict-affected areas. Strategies to address these themes can be guided by the GRISP areas targeting the mobilization of people (through engaging communities) and maximizing reach (through designing services to effectively deliver vaccines to all target groups and improve equity), respectively [16].

Lastly, there are opportunities for strategy development at the JTHI steps for which we found fewer strategies (Intent, Experience of Care, and After-Service). Immunization intent is related to the caregiver’s decision-making power and self-efficacy, which is influenced by societal gender norms and roles [15]. Although women are traditionally the primary caregivers in households, they often lack the decision-making power to influence the health of their children [25,26,27]. Moreover, the link between maternal education and child immunization has been thoroughly documented in the literature and can be an important contributor to the “social distance” between caregivers and health workers, which is a key component of addressing the experience of care [4,7,15]. An emphasis on improving maternal health literacy can be an important consideration to address this gap [25,26,27]. Further, an ecological framework can be useful in identifying gaps and interventions targeting individual and household factors affecting women’s decision-making process, including health literacy and capacity for negotiation within households and within healthcare settings [25].

Given that step 6 feeds back into the JTHI cycle and can influence other dimensions of the care-seeking process, more attention should be brought to assessing the opportunities at that step. JTHI step 6 strategies focus on clients’ access to information about the after-effects following immunizations (AEFI), knowledge about follow-up appointments and when to return for services [15]. At that step, very few strategies targeted performance rewards and processes for evaluating programs (taking stock of post-care successes and failures). Opportunities for strategy development addressing these themes can be informed by two of the GRISP areas of action. Firstly, in the area of maximizing reach, there are opportunities for transformational investments related to building the capacity of vaccinators, including the provision of performance rewards for healthcare workers [16]. Secondly, in the area of monitoring progress, there are opportunities to develop strategies related to the evaluation of programs through surveys and review, which includes taking stock of post-care successes and failures [16].

### 4.1. Strengths and Limitations of the Mapping

Our synthesis builds on and reinforces previous work to describe pro-equity strategies for immunizations in Gavi-supported countries (13). Moreover, we demonstrate the utility of using JA reports to characterize and understand country-level practices to improve immunization coverage and equity. The mapping of strategies to multiple key dimensions and themes, including JTHI steps, systemic factors emphasized by GRISP and relevance to key populations, allowed us to create a searchable database which can be used for knowledge sharing among countries, as well as inspiration for addressing gaps and developing targeted approaches [17].

An important limitation of the mapping is that it considers only the strategies that are being reported by countries in the JA and MSD reports. Therefore, the data extracted from these reports might not include all strategies used at the national or subnational levels. Additionally, data might be incomplete for the countries where only one or two data points were available. Moreover, the review represents a snapshot of the strategies that are currently being implemented in all countries, without evaluating the implementation process or the efficacy of these strategies directly. Therefore, the recommendations that can be made from the data are limited to “what” is being done, without being able to explain “how” these strategies are implemented or how well they worked. This is an important evidence gap that needs attention.

### 4.2. Implications for Policy and Practice

An equity-driven approach is necessary to identify immunization gaps and opportunities to reach underimmunized and zero-dose children in LMICs. The awareness of current strategies and gaps can inform equity-driven immunization programs, policies and decision making targeting zero-dose and underimmunized children in LMICs. The dissemination of these findings can also promote knowledge and expertise sharing between countries and can serve as a base for strategic planning and scale-up of interventions promoting equitable immunization programs, particularly between countries in similar socio-political or cultural contexts. Future directions for this work could include an analysis of the country-level impact of these strategies on immunization inequities. An emphasis on monitoring and evaluation activities could further characterize the areas where investments are justified and desired.

Given that Gavi-supported countries submit JA or MSD reports yearly, it would be beneficial to continuously update the data and searchable database on a yearly basis, as reports become available [17]. Moreover, given the disruption of services that occurred due to the COVID-19 pandemic, we recommend a detailed analysis of the pro-equity mitigation strategies deployed in 2020 since the start of the pandemic. Lastly, continued work to categorize strategies based on key dimensions, such as relevance to key populations (urban, remote rural and conflict) and gender considerations, can provide opportunities for more targeted solutions to implementation challenges.

## 5. Conclusions

We conducted a targeted mapping and synthesis of pro-equity strategies for immunization in 61 Gavi-supported countries. Strategies were thematically mapped against the JTHI steps and the GRISP framework, which allowed us to identify common practices and opportunities for future investments. Findings have been incorporated into a learning tool and searchable database of pro-equity strategies, which can serve as a resource and a guide to other countries who want to improve their immunization coverage and equity [17].

## Figures and Tables

**Figure 1 vaccines-11-00191-f001:**
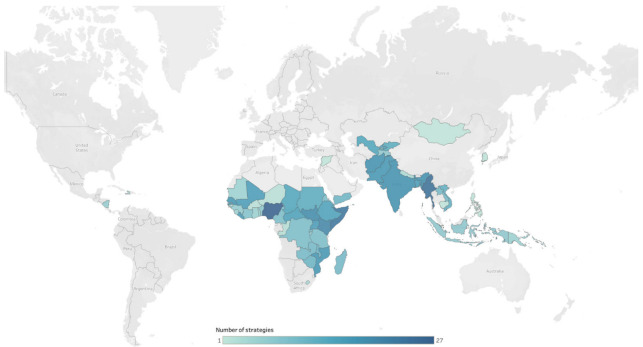
Number of strategies per country.

**Figure 2 vaccines-11-00191-f002:**
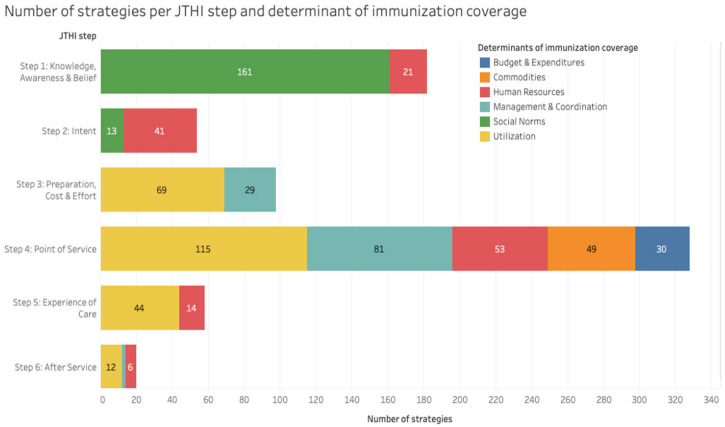
Relative proportion of strategies per JTHI step and determinant of immunization coverage.

**Figure 3 vaccines-11-00191-f003:**
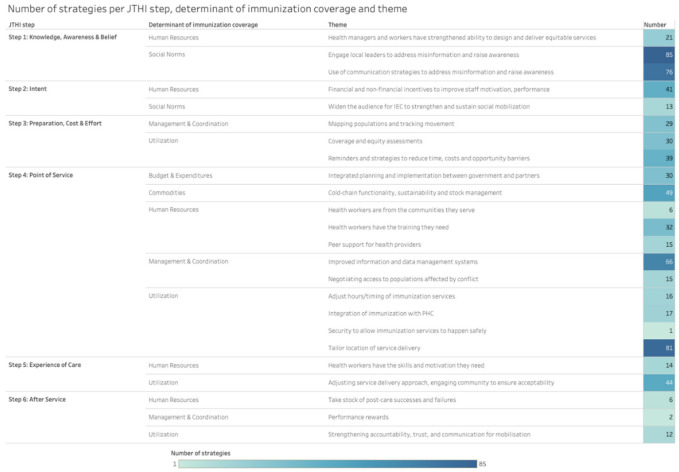
Number of strategies per JTHI step, determinant of immunization coverage and theme.

**Table 1 vaccines-11-00191-t001:** Key caregiver and health worker dimensions of the JTHI framework [15].

JTHI Step	Caregiver	Health Worker
1: Knowledge, Awareness and Beliefs	Practical knowledge, norms and values, trust in vaccines and providers	Practical competencies, norms and values, perception of clients
2: Intent	Decision-making power, self-efficacy	Motivation/satisfaction, social recognition, community respect
3: Preparation, Cost and Effort	Logistics of remembering, transport, childcare, juggling competing priorities, social and opportunity costs	Preparing getting to clinic/outreach site, opportunity costs
4: Point of Service	Appropriateness and convenience of services, service hours, social distance	Training, job aids, workload, facility/flow
5: Experience of Care	IPC and treatment by health workers, physical conditions, use of home-based records, client satisfaction	Interpersonal communication skills, trust building, pain mitigation, training and experience, social distance
6: After-Service	Information on AEFI and when and where to return, sharing +/− experience with community, reinforcement of vaccination as norm	Family and community respect, celebration of achievements, supportive supervision

## Data Availability

The complete dataset for this study is available online in the UNICEF Immunization, Gender and Equity Solutions Library [17]. Data sources (Joint Appraisal and Multi-Stakeholder reports) can be found on the Gavi, the Vaccine Alliance website.

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
