# Peer review of "Advancing Immunization Coverage and Equity: A Structured Synthesis of Pro-Equity Strategies in 61 Gavi-Supported Countries"

_vaccines, 2023, doi:10.3390/vaccines11010191_

Round 1
Reviewer 1 Report
This is a well written, very interesting paper on a topic of high relevance. The methodology and the results are clearly presented as well as the study’s strengths and limitations. I just have a few minor comments for further clarification and for focusing the discussion on a few key topics that would also be relevant to a wider audience:
- Figure A1 has a very low resolution and it is hard to read. It might be better presented as a table.
- Figures B1 and B2 would also be better presented and more readable in a Table format, and the color code can be kept.
- Figure 3. The last column lacks a name. And the color codes should be redefined for this figure as well.
- Line 256-260: Vaccine hesitancy is a really important topic that is not addressed enough in most settings. This is also true for the anti-vaccine movement (see PMID 35096665). The discussion could benefit from more information regarding to which extent this is an important factor in the surveyed countries.
- Line 277: The discussion could explore to what extent these proposed improvement are feasible in each setting.
- Line 295: Lack of knowledge about vaccines has been indeed associated with decreased vaccine utilization. This can and should be easily addressed, an interesting publication has shown the role of engaging general practitioners as well as pediatricians in discussion with parents regarding vaccines, see PMID 35335036.
- The conclusions could include one or two important directions for actionable advice derived from this publication.
- The authors should check whether all named persons in the acknowledgement section have given their consent to have their names published.
Reviewer 2 Report
This is a very interesting report, well presented. My only comment is that it is written for the initiated. It is best understood if read with the Dadari report or if one is familiar with that report. A list of abbreviations (similar to that in the first page of the Dadari paper would be helpful. Some points would be helpful if explained; for example 'multi-stakeholder dialogues' - are they interviews? questionnaires?
Minor point: line 197 'step 5 (point of care) should it be 'experience of care' so as not to confuse with step 4?
Reviewer 3 Report
The study aimed to identify strategies to improve equity in immunization among low and middle-income countries, supported by GAVI. The idea is not new, in 2021 a first phase results were published concerning 13 countries. Now additional 48 countries were included and new equity filters were added. The authors made good work to identify and map various actions undertaken to minimize inequities in immunization. The methods and results are well described and conclusions are adequate.
The main limitation of the study is that pro-equity strategies are "only" identified, there is no data about their implementation, as it was underlined by the authors. Nevertheless, knowledge about those strategies is valuable and can help to develop targeted approaches. The authors declared that a searchable database was created, it would be worth to mention where it could be found.
Author Response
Response to Reviewer 3 Comments
Point 1: The authors declared that a searchable database was created, it would be worth to mention where it could be found.
Response 1: The searchable database can be found under the following reference #17 linked at the end of the manuscript in the “References” section. The reference number has been added throughout the revised manuscript whenever the database is mentioned, to facilitate access.
UNICEF. Immunization, gender and equity: Solutions library. 2022. Available online: https://www.ige.health/solutions/ (accessed on 18 December 2022).